# The New Coronavirus Infection (COVID-19) and Hearing Function in Adults

**Maria Y. Boboshko [1],\*, Ekaterina S. Garbaruk [1] , Sof'ya M. Vikhnina [1], Larisa E. Golovanova [2], Elena A. Ogorodnikova [3], Anna V. Rabchevskaya [1] and Ekaterina V. Zhilinskaia [4]**

[1] Laboratory of Hearing and Speech, Pavlov First St. Petersburg State Medical University, Str. L'vaTolstogo 6-8, 191022 St. Petersburg, Russia; kgarbaruk@mail.ru (E.S.G.); sofya.vikhnina@mail.ru (S.M.V.); rabchevskaya.anna96@gmail.com (A.V.R.)

[2] Municipal Audiology Department, Saint-Petersburg Geriatric Medico-Social Center, Rigskii pr. 21, 190103 St. Petersburg, Russia; lgolovanova@inbox.ru

[3] Laboratory of Speech Psychophysiology, Pavlov Institute of Physiology RAS, Emb. Makarova 21, 199034 St. Petersburg, Russia; elena-ogo@mail.ru

[4] Department of Otorhinolaryngology and Ophthalmology, St. Petersburg University, 21 Line V.O., 199034 St. Petersburg, Russia; xelloss@mail.ru

\* Correspondence: boboshkom@gmail.com; Tel.: +7-812-338-60-34

**Abstract:** In this study, we assessed the impact of COVID-19 on the hearing function in adults. A total of 161 subjects were examined, and the results of a previous audiological examination of 24 patients were reviewed. Pure tone audiometry, impedancemetry, speech audiometry in quiet and noise, the Binaural Fusion Test, the dichotic digits test, and a cognitive status examination were performed. A total of 81% of patients complained about hearing disorders, and 43% noted memory impairment. According to pure tone audiometry, 24% of the subjects had normal hearing, while 76% had some degree of hearing loss. No significant changes in hearing thresholds were found in comparison with audiological examinations performed before COVID-19. Disorder of monosyllabic words' intelligibility in quiet was found in 33% of patients, and in 42% in noise, along with low indicators in the dichotic digits test in 54% of patients. Moreover, 71% of patients had low scores on the MoCA scale that indicated cognitive impairment. Conclusions: The deterioration of speech test scores in patients after COVID-19 can occur due to central auditory processing disorders (CAPD), memory impairment, or changes in cognitive status in general.

**Keywords:** COVID-19; hearing; speech intelligibility; Russian matrix sentence test; dichotic digits test; CAPD; Montreal Cognitive Assessment

## 1. Introduction

Since 2020, the pandemic of the new coronavirus (COVID-19) has afflicted the world. As of April 2022, the pandemic has caused about 400 million confirmed cases and more than fivemillion deaths worldwide. The virus can damage the lungs, heart, neural system, etc., which increases the risk of long-term health problems. Clinical manifestations, pathophysiology, disease prognosis, and long-term consequences of COVID-19 infection are actively being studied worldwide.

COVID-19 is a highly contagious infectious disease caused by the SARS-CoV-2 RNA virus. The virus penetrates the epithelium of the upper respiratory and gastrointestinal tracts. These are the following routes of COVID-19 transmission: aerosol transmission, airborne, and contact. SARS-CoV-2 targets various systems and organs, primarily type II alveolar lung cells. Cardiovascular, immune, hematological, skeletomuscular, gastrointestinal, hepatic, renal, and peripheral and central nervous systems may also be affected. The neurotrophiceffect of the virus is caused by its ability to penetrate cells with ACE2 receptors, including the cells of the brain and pituitary gland. Penetration of SARS-CoV-2 hematogenously or through the lamina cribrosa can lead to brain damage [1,2].

Some viral infections are known to cause hearing loss. The pathogenesis of hearing loss has been described by different researchers. Auditory deficits can be attributed to the damage tothe cochlea or the involvement of the central auditory pathways [3]. Hearing loss, dizziness, and tinnitus are described among audio-vestibular disorders in patients during or after COVID-19. A large number of papers are currently available that are devoted to the state of the auditory and vestibular pathways in those who recovered from COVID-19, including clinical cases, prospective and retrospective studies, and systematic reviews.

Generally, the impaired auditory function is considered to be a relatively rare manifestation of COVID-19. However, since the pandemic's onset, cases of sudden sensorineural hearing loss (SNHL) up to deafness against the background of COVID-19 have been described in both men and women of different age groups. Hearing loss has occurred not only due to severe manifestations of the disease, but even in mild cases. Both ears may be affected [4,5]. In some patients, complaints about hearing loss were accompanied by tinnitus [6]. In some subjects, COVID-19 was the cause of deafness, requiring cochlear implantation [7].

Numerous investigations have been performed in order to obtain systematic data on the impact of COVID-19 on auditory function. These studies were conducted in patients both with hearing loss complaints and without them. Research methods usually included otoscopy, tympanometry, pure tone audiometry (PTA), and extended high-frequency audiometry; in some cases, they were supplemented by otoacoustic emission (TEOAE, DPOAE) and auditory brainstem response (ABR) registrations [8–10].

In a large number of studies, there were no significant changes found in patients who had no complaints about hearing loss, and there were no significant changes in the PTA results of patients after COVID-19 compared to the control group or compared to their hearing thresholds registered before the disease [8,11]. However, increased hearing thresholds and decreased amplitude of OAEs in patients after COVID-19 compared to the control group (especially in the high-frequency region) were described in some papers [9,12].

Audiological manifestations related to SARS-CoV-2 infection are varied. However, the following patterns were found among patients with COVID-19-induced sudden SNHL: age over 30 years, two ears impaired, symmetric hearing loss, complicated with tinnitus [10]. Some patients with normal hearing complained of tinnitus [13]. Numerous researchers performed studies to analyze the incidence of audio-vestibular disorders during COVID-19. They compared the number of clinical cases for sudden SNHL, tinnitus, facial nerve paresis, and vertigo during the pandemic year to those in the year prior to the pandemic. Most scientists have found that the incidence of coronavirus-associated SNHL does not differ significantly in comparison with previous years [14]. However, V. Fidan et al. (2021) described an increase in the number of SNHL cases during the pandemic [15]. Generally, the studies have shown a wide range of results (including the occurrence rate of hearing loss related to COVID-19) [16].

The pathogenesis of COVID-19-associated hearing disorders includes viral inflammation of the inner ear, autoimmune disorders caused by a cross-reaction of antibodies or T-cells to inner ear antigens, cochleo-vestibular damage secondary to cytokine storm, and blood coagulation lesions that may lead to thromboembolic complications of the auditory and vestibular analyzers and ischemia [17,18]. It is suggested that SARS-CoV-2, which has neuroinvasive properties, can cause central nervous system disorders, affecting higher mental functions and leading to memory impairment and neurocognitive problems [19–21]. Complaints about memory impairment are widespread among those who recovered from COVID-19 [22].

Scientific data concerning hearing status after COVID-19 are scattered. Despite the possibility of central auditory processing disorders (CAPD), the impact of COVID-19 on the central part of the auditory system is almost unevaluated. There is also only a small number of studies aimed at the evaluation of speech intelligibility in patients after COVID-19 [20]. The scientific data and experience accumulation by otorhinolaryngologists and audiologists

will allow for more effective rehabilitation for patients complaining about hearing loss after COVID-19.

The aim of this study was to evaluate the impact of COVID-19 on peripheral and central parts of the auditory system in adults.

## 2. Materials and Methods

### 2.1. Patients Studied

The inclusion criteria of this study were (1) age above 18 years, (2) complaints of hearing impairment and/or tinnitus during or after COVID-19, and (3) recovery from COVID-19 no earlier than 2 weeks and no later than 24 weeks before the audiological check-up. The exclusion criteria were (1) age below 18 years, (2) the absence of hearing impairment complaints, and (3) comorbidity (e.g., severe somatic and neurological pathology).

Between 29 April 2021 and 9 November 2021, 161 individuals (322 ears) who complained about hearing loss, speech intelligibility disorder, and/or tinnitus after COVID-19 of varying severity were examined. The sample comprised 120 women and 41 men from 22 to 92 years of age (60 years, SD = 13.1). Among them, 137 (85%) were primary patients of audiologists, and 24 patients (15%) had been previously checked by audiologists and had results of previous audiological tests no more than 12 months prior to having COVID-19. A comparison of the complaints and the results of an audiological examination (PTA, impedancemetry) before and after COVID-19 was performed in the group of 24 patients (48 ears); extended speech testing was also conducted in this group.

### 2.2. Measures

In addition to the complaints survey, medical history, and otorhinolaryngological examination, all the patients underwent a basic audiological examination, which included PTA for air (0.125, 0.25, 0.5, 1, 2, 4, and 8 kHz) and bone conduction (0.25, 0.5, 1, 2, 4, and 8 kHz). The degree of hearing loss was assessed in accordance with ASHA recommendations [23], based on the average air conduction thresholds at four frequencies (mean $PTA_{0.5, 1, 2, 4}$): normal, $\leq$25 dB; mild, 26–40 dB; moderate, 41–55 dB; moderately severe, 56–70 dB; severe, 71–90dB.Impedancemetry (tympanometry and ipsilateral acoustic reflexmeasurement at 500, 1000, 2000, and 4000 Hz) and speech audiometry in quiet were also performed for all 161 patients. The maximum speech recognition score in quiet ($MSRS_Q$) was assessed by presenting monosyllabic words (20 words in the test list) for each ear separately.

The central auditory processing assessment included the Random Gap Detection Test, the maximum speech recognition score of monosyllabic words in noise ($MSRS_N$) evaluation, and the dichotic digits test were performed for all 161 patients, and the Binaural Fusion Test and the Russian matrix sentence test in quiet and noise were additionally performed for 24 patients.

The Random Gap Detection Test (RGDT)with R. Keith's (2000) modification was used [24]. Pure tones (0.5, 1, 2, and 4 kHz) were presented with brief silent intervals through headphones at a comfortable sound level binaurally. The duration of one acoustic signal was 15 ms. Signals with silent intervals of duration from 0 to 40 ms were run in a random order. The task of the listener was to answer whether he/she perceived the signal as one sound or two. A total of 9 signals at each frequency were presented in the test, and then the minimum silent interval that the listener picked up at that frequency was estimated (the listener distinguished two sounds in the presented signal). The gap detection threshold was calculated as the average of the thresholds obtained at the studied frequencies. Before the test was run, the patient was provided with a training track to clarify whether he/she had understood the task correctly. Normally, the gap detection thresholds at the 3 frequencies should be less than 20 ms. The test is considered to be failed if the gap detection threshold exceeds 40 ms for two or more frequencies or if all paired signals are perceived as single, and, conversely, if single signals appear to be paired.

The maximum speech recognition score in noise ($MSRS_N$) was assessed by presenting recordings of single words (20 words in the test list) to each ear separately with a white noise background, and signal-to-noise ratio (SNR) = 0 dB.

In the dichotic digits test (DDT), two different pairs of two-digit numbers from 11 to 99 were presented simultaneously to different ears at a comfortable sound level. A total of 20 pairs of numbers were given. The patient's task was to repeat both numbers heard in random order. The result was calculated as the percentage of the pairs repeated correctly. People with normal hearing should have scores ≥90%, and people with mild to moderate SNHL should have scores ≥80% [25].

In the Binaural Fusion Test (BFT), 20 monosyllabic words were each divided into two halves and presented to the patient via headphones at a comfortable level: one-half of the word was presented to one ear, and the other half was presented to the other ear immediately thereafter. The patient's task was to repeat each word that was heard. The result was calculated as the percentage of the words repeated correctly. Normally, the percentage of intelligibility in this test may be lower than the percentage of monaural intelligibility of single words ($MSRS_Q$), but this difference (BFT, Δ) should not exceed 20%.

In the Russian matrix sentence test (RuMatrix), speech material is represented by sentences of 5 words: the 1st word is a name, the 2nd is a verb, the 3rd is a numeral, the 4th is an adjective, and the 5th is a noun. For example, "Ivan wants five red halls" [26,27]. The patient's task was to repeat the sentence heard, or at least its individual words. During testing, 20 sentences in quiet with an adaptive procedure were presented monaurally. We assessed speech recognition threshold in quiet ($SRT_Q$) in dBSPL. Then, the patients were tested in background noise: two tracks of 20 sentences were presented to each ear with an adaptive procedure with a fixed noise level of 65 dB SPL. The intensity of the speech signal changed automatically, decreasing when the subject answered correctly and increasing in case of an incorrect answer. We estimated the speech recognition threshold in noise ($SRT_N$) in dB SNR. The measurements were performed in an open-set format (subjects repeated overheard sentences without any visual support). The first track was considered to be a training one.

The following equipment was used for pure tone audiometry, speech audiometry, and non-verbal psychoacoustic tests: an MA 42 clinical audiometer (Germany), TDH39 headphones, an AEG portable MP3 CD player, and a CD with non-verbal and speech tests. A laptop with Oldenburg Measurement Application software (HörTech GmbH, Oldenburg, Germany), an EarBox sound card (Auritec, Hamburg, Germany), and Sennheiser HDA200 headphones were used for the RuMatrix.

The Montreal Cognitive Assessment (MoCA) was used to evaluate cognitive status [28].

*2.3. Statistical Analysis*

For the statistical analysis of distribution parameters, standard indicators were used: sample size, sample mean, standard deviation, standard error of the mean, and 95% confidence interval (CI). Wilcoxon's T-test for related samples was used to assess the reliability of differences. Spearman's rank correlation coefficient was used to evaluate interconnections.

## 3. Results

In the subjects tested (*n* = 161), 81% (130 patients) complained about hearing impairment (manifestation or deterioration of hearing loss, decreased speech intelligibility), 32% (52 patients) were bothered by tinnitus, and 43% (69 patients) noted memory impairment. The results of the examination of these patients are presented below.

*3.1. Pure Tone Audiometry*

According to PTA results (Figure 1), 24% of the patients had symmetric normal hearing (39 patients, 78 ears) and 76% had unilateral or bilateral hearing loss (122 patients) from mild to moderately severe; the hearing loss was unilateral in 28 patients. Thus, there were 33% (106) normal hearing ears in the group (78, bilateral normal; 28, unilateral), and 67%

(216) hearing-impaired ears (188, bilateral hearing loss; 28, unilateral). In 84% (181 ears), there was SNHL, and in 16% (35 ears), there was a mixed hearing loss that occurred due to either adhesive otitis (32 ears) or otitis media with perforation (3 ears).

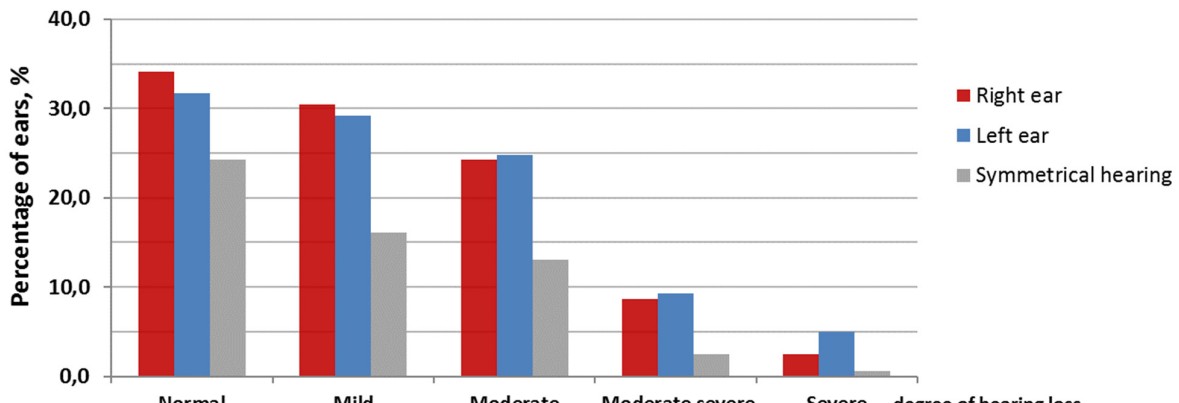

**Figure 1.** Distribution of average hearing thresholds at 0.5, 1, 2, and 4 kHz ($PTA_{0.5, 1, 2, 4}$) for the right and left ears in patients with normal hearing and with hearing loss of different degrees.

### 3.2. Impedancemetry

A type "A" tympanogram was registered bilaterally in 69% of measurements (112 patients, 224 ears) and unilaterally in 4% of measurements (14 patients, 14 ears). Other types of tympanograms were obtained less frequently: type "As" in 8% (15 patients, 25 ears), "Ad" in 6% (13 patients, 20 ears), "B" in 5% (11 patients, 16 ears), and "C" in 6% (14 patients, 20 ears). In 2% (three patients, three ears), there was a perforation of the tympanic membrane.

The results of acoustic reflexometry varied significantly. At normal hearing thresholds (106 ears), acoustic reflex due to ipsilateral stimulation was recorded at all speech frequencies in both ears in 61% of cases (65 ears), and in one of the ears in 15% of cases (16 ears); it was not recorded in both ears in 7% of cases (7 ears), and it was not recorded in one of the ears in 8.5% of cases (9 ears). Of 216 hearing-impaired ears, it was registered in both ears in 22% (47 ears) and in one of the ears in 7% (15 ears); it was not registered in both ears in 19% (41 ears) or in one of the ears in 10% (21 ears).

### 3.3. The Random Gap Detection Test

Of 39 patients with normal hearing, 23 (60%) successfully managed with the RGDT. The average detection threshold registered was $7.4 \pm 0.62$ ms, and only 21 (17%) of 122 patients with hearing loss successfully completed the test, with an average threshold of $9.6 \pm 0.61$ ms.

### 3.4. Speech Audiometry in Quiet

Low $MSRS_Q$ was revealed in 33% of patients (their recognition did not exceed 60%). For patients with normal hearing, the mean $MSRS_Q$ was $82.7 \pm 1.61\%$, and for those with hearing loss, it was $70.6 \pm 1.35\%$.

### 3.5. Speech Audiometry in Noise

The $MSRS_N$ was low in 42% of the patients. The group average $MSRS_N$ was $77.8 \pm 2.2\%$ for ears with a normal threshold and $65.5 \pm 2.9\%$ for ears with hearing loss.

### 3.6. Dichotic Digits Test

Low indicators in the DDT were revealed in 54% of patients. It averaged $73.8 \pm 1.94\%$ for the patients with normal hearing and $56.7 \pm 2.21\%$ for those with hearing loss.

### 3.7. The MoCA

In 114 (71%) patients, the MoCA scale values were less than 26 points (19–24 points), indicating cognitive impairment.

For additional analysis, 24 patients aged 22 to 85 years (mean age 58 years, SD = 16.7) with previous check-up results were identified. Patients in the group underwent extended speech testing after COVID-19, including the BFT and RuMatrixtests.

### 3.8. Extended Examination Results of 24 Patients Checked by Audiologists before COVID-19

Subjective complaints of the patients in this group were compared with those before COVID-19 (based on analysis of outpatient cards) and after recovery from COVID-19 (Table 1).

**Table 1.** Subjective hearing complaints in patients before and after COVID-19.

| Number (%) of Patients in the Group (*n* = 24) | | | | |
|---|---|---|---|---|
| Before COVID-19 | After COVID-19 | | | |
| Complaints about hearing loss, poor speech intelligibility, tinnitus | Deterioration and occurrence of new complaints | Of them: | | |
| | | Deterioration of hearing and speech intelligibility | Tinnitus | Dizziness and noise "in the head" |
| 14 (58%) | 11 (46%) | 7 (64%) | 3 (27%) | 1 (9%) |

Additionally, nine people (38%) complained about impaired concentration and memory lesionsafter recovery from COVID-19.

In 24 patients who had been checked by audiologists before COVID-19, 16 (67%) had hearing loss: in 58% of cases (14 individuals, 23 ears), it was SNHL, and in 8% of cases (2 individuals, 3 ears), it was mixed. Normal hearing was registered in 33% of patients (8 individuals, 15 ears), and six of them had normal hearing. In the patients after COVID-19, the contribution of SNHL increased to 75% (2 people, 2 ears), and the proportion of normal hearing decreased to 25% (7 people, 12 ears, 5 patients with bilaterallynormal hearing). No significant changes in mean $PTA_{0.5, 1, 2, 4}$ were found in the group (Wilcoxon T-test for related samples). However, there was a predominant tendency observed of its increase, on average for the group, by $3.3 \pm 0.41$ dB (right ear) and $3.1 \pm 0.46$ dB (left ear). This tendency was also evident in the distribution of mean $PTA_{0.5, 1, 2, 4}$ (Figure 2).

A decrease in the proportion of bilateral "A" tympanogram by 12.5% (THREE patients registered a "C" type tympanogram after the disease) can be also related to the manifestation of a negative trend after the disease, in comparison with the examination before COVID-19. Acoustic reflexometry results also deteriorated in these patients—there was a narrowing of the registration frequency range due to the absence of an acoustic reflex at the frequencies 2 and 4 kHz. In cases of normal hearing (15 ears), the acoustic reflex was registered at all frequencies in 60% (nine ears) and was not registered at any frequency in one ear. In cases of hearing loss (33 ears), the acoustic reflex was recorded at all frequencies in 15% (five ears) and was not recorded at any frequency in 48% (16 ears).

Only 10 (42%) of 24 patients (eight with normal hearing and two with mild hearing loss) successfully completed the gap detection test; the other 14 patients (58%) failed it.

The results of the extended speech testing of the patients in this group after COVID-19 are presented in Table 2. The patients differed in age and hearing status (mean age of normal-hearing patients was $53.4 \pm 4.3$ years, and the mean age of hearing loss patients was $61.5 \pm 2.8$ years); consequently, normative values for the RuMatrix test in quiet and in noise are presented as a range from young to senior age.

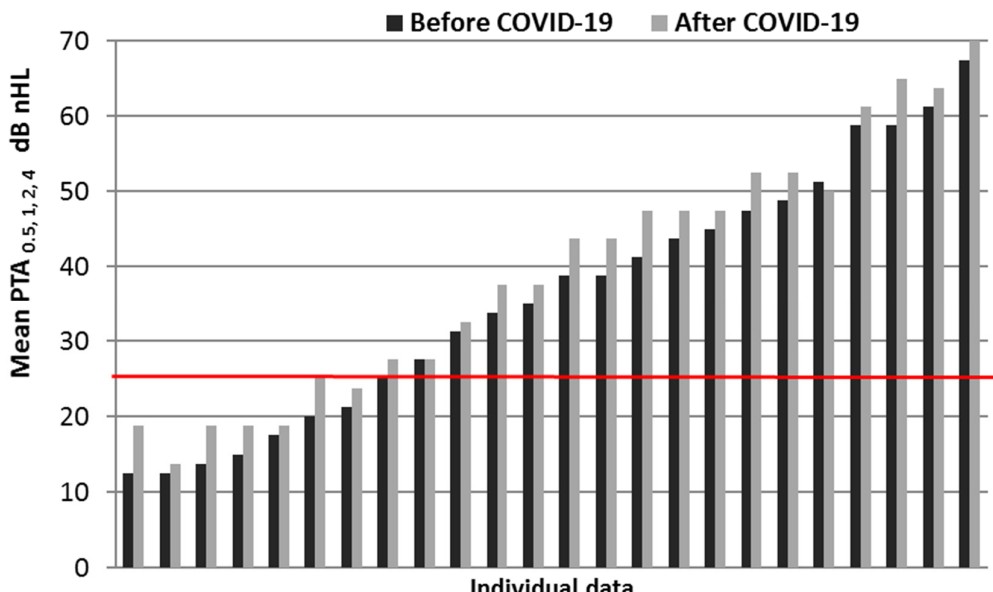

**Figure 2.** Individual PTA data for the right ear (air conduction) before and after COVID-19. The boundary line of "normal" hearing (mean $PTA_{0.5, 1, 2, 4} = 25$ dB) is highlighted.

**Table 2.** Results of extended speech testing of patients after COVID-19 ($n$ = 24, 48 ears).

| Tests | Measurement Results M ± m (CI$_{95\%}$) | | | | Normative Values | |
|---|---|---|---|---|---|---|
| | Average Values (per Group) | | Average Values (per Hearing Function) | | | |
| | AD (24 Ears) | AS (24 Ears) | Normal ($n$ = 8) | Hearing Loss ($n$ = 16) | Normal | Hearing Loss |
| MSRS$_Q$ (%) | 82.1 ± 3.2 | 81.2 ± 4.6 | 92.2 ± 2.1 (CI = 4.1) | 73.8 ± 3.7 (CI = 7.3) | ≥90 | ≥80 |
| MSRS$_N$ (%) | 65.0 ± 3.8 | 62.5 ± 3.7 | 76.3 ± 2.4 (CI = 4.8) | 57.5 ± 3.3 (CI = 6.4) | ≥70 | ≥60 |
| SRT$_Q$ (dB SPL) | 37.5 ± 4.6 | 38.4 ± 4.2 | 27.3 ± 2.6 (CI = 5.1) | 62.9 ± 2.9 (CI = 5.8) | 13.3 ÷ 36.2 * | 36.8 ÷ 83.0 * |
| SRT$_N$ (dB SNR) | −3.7 ± 1.4 | −3.9 ± 1.2 | −6.9 ± 0.4 (CI = 0.7) | 3.7 ± 1.0 (CI = 1.9) | −8.7 ÷ −6.9 * | −3.7 ÷ 0.5 * |
| DDT (%) | 61.7 ± 4.6 | | 68.0 ± 4.9 (CI = 9.7) | 55.9 ± 4.3 (CI = 8.5) | ≥90 | ≥80 |
| BFT (Δ, %) | 6.4 ± 3.1 | | 5.7 ± 3.9 | 7.1 ± 4.2 | ≤20 | ≤20 |

M ± m, arithmetic mean ± standard error of the mean; CI$_{95\%}$, 95% confidence interval; * range of normative indicators from young to old age: adapted from [27].

The mean (M ± m) data of the MoCA cognitive test in the group were less than normal, with a score of 23.4 ± 0.8. Individual scores ranged from 15 to 29, and in 15 patients (63%), they were less than 26 (below normal). A mean MoCA test score of 25 ± 0.95 was observed in patients with normal hearing, and in those with hearing loss, it was even lower (21.5 ± 1.21), but the difference did not reach the level of significance.

The correlation analysis shows that the crucial factors that most impacted the results are the duration and severity of COVID-19. Duration of the disease was found to correlate moderately with the mean $PTA_{0.5, 1, 2, 4}$ and with the average hearing thresholds in the range of 4 ÷ 8 kHz ($0.2 < p < 0.1$). A moderate negative correlation was revealed between the cognitive test MoCA score and the duration of the disease ($0.1 < p < 0.05$), and a mild correlation was observed between the MoCA score and severity of the disease ($p < 0.2$).

A strong correlation between the RuMatrix test results in noise and duration of the disease was also observed ($0.05 < p < 0.02$); the correlation between the RuMatrix test results in noise and disease severity was moderate ($0.1 < p < 0.05$). For the RuMatrix test results in quiet, this correlation was shown to be moderate for both disease duration ($0.1 < p < 0.05$) and severity of COVID-19 ($0.2 < p < 0.1$). In addition, a moderate correlation between the RuMatrix scores and the MoCA scores was found for recognition in noise ($0.1 < p < 0.05$) and in quiet ($p < 0.2$). A moderate correlation between the MoCA scores and DDT was also revealed ($0.1 < p < 0.05$).

In general, the correlations found are consistent with the hypothesis that COVID-19 may lead to the impairment of central auditory pathways. Verified deterioration of psychoacoustic test results (RGDT, speech tests, especially when performed in noise) and cognitive testing results may be determined by this impact. The correlations with the RuMatrix sentence test indicate a specific impact of the virus on memory function.

## 4. Discussion

In contrast to other studies dedicated to hearing assessment after COVID-19, in which PTA results of listeners without any complaints about hearing and without any auditory disorders in their history were analyzed [12], in this study, we included only patients who had complaints about auditory disorders. Special emphasis should be placed on the group of 24 patients (48 ears) who had been previously checked by audiologists, and the results of previous audiological tests (PTA, impedancemetry) registered in their outpatient medical charts. The complaints analysis shows that after the disease, 11 patients (46%) noted a deterioration of their complaints and the development of new ones. The majority of patients were bothered by progressive hearing loss and impaired speech recognition; nine patients complained about decreased concentration and memory impairment. PTA results analysis did not reveal any significant hearing threshold changes at all frequencies in 24 patients who had been examined before COVID-19. Nevertheless, the mean $PTA_{0.5, 1, 2, 4}$ tended to increase on average by $3.3 \pm 0.41$ dB (right ear) and $3.1 \pm 0.46$ dB (left ear). A moderate correlation ($0.2 < p < 0.1$) was found between the duration of the disease and the mean $PTA_{0.5, 1, 2, 4}$, and also the mean hearing thresholds at 4 and 8 kHz. The acoustic reflex measures were found to be deteriorated after COVID-19—the acoustic reflex was registered at all frequencies in only 60% of ears with normal hearing and 15% of hearing-impaired ears, which may be due to both middle ear dysfunction after the disease and the influence of SARS-CoV-2 on the central nervous system.

To evaluate the temporal resolution of the human auditory system, the RGDT was performed. Its results depend mostly on the functioning of the central auditory pathways. The auditory system's ability to discriminate subtle temporal aspects of sound signals is the basis of speech comprehension. The RGDT is sensitive to cortical disorders, especially of those in the left hemisphere; one of the advantages of the RGDT is the little dependence of the results on a patient's linguistic skills and training [25]. Poor results of the RGDT in many patients were noted in our study: only 23 (59%) of 39 patients with normal hearing and 21 (17%) of 122 patients with hearing loss accomplished the RGDT successfully. The interconnection between poor RGDT results and deteriorated cognitive abilities after COVID-19 was proposed in this study, and confirmed by poor MoCA results in 114 (71%) patients. The data achieved by other researchers also correspond with these findings [29]. Scientific data concerning the correlation between temporal resolution and cognitive functions are controversial. While some studies have shown the absence of such correlation [30], others, on the contrary, have revealed the interconnection between information processing speed, as well as cognitive function, and temporal resolution [25].

Although speech audiometry in quiet does not relate to the tests for CAPD detection, low scores of speech intelligibility in quiet should be taken into account. In this study, the low $MSRS_Q$ ($\leq 60\%$) was revealed in 53 (33%) of 161 patients due to the evaluation of monosyllabic words' intelligibility. The RuMatrix sentence test in quiet showed the normal range of $SRT_Q$ according to the age and hearing thresholds in all 24 patients who underwent

this test [27]. It might be determined by the redundancy of speech material in the RuMatrix test in comparison to the monosyllabic test. A high correlation ($p < 0.01$) between $SRT_Q$ and mean $PTA_{0.5, 1, 2, 4}$ was observed, which confirms that speech discrimination in quiet depends mainly on the peripheral hearing functioning, as previously shown in other investigations [27,31].

Speech audiometry in noise was performed via the monaural presentation of monosyllabic words ($n = 161$, 322 ears) and the RuMatrix test ($n = 24$, 48 ears). The low $MSRS_N$ ($\leq 60\%$) was revealed in 68 (42%) of 161 patients. Based on the RuMatrix in noise results, $SRT_N$ fell short of the normative range in 4% (two ears) of patients with normal hearing and in 25% (eight ears) of patients with hearing loss. In one patient with moderately severe hearing loss, $SRT_N$ reached +5.3 (right ear) and +9.1 (left ear) dB SNR. These results comply with the findings of other researchers, who proved that good speech discrimination in quiet is not a predictor of good speech discrimination in noise [32]. As it has been shown in several clinical studies, testing with speech in noise is sensitive to the auditory cortex pathology, including its degenerative age-related changes. The deficit of the ear contralateral to the dysfunctioning auditory cortex was found in those tests [33]. Poor results, although to a lesser extent, were also obtained in patients with brainstem pathology [34]. The revealed correlation between the RuMatrix in noise results and the duration and severity of COVID-19 allows us to suggest the influence of the virus on the auditory cortex.

The Binaural Fusion Test (BFT) was performed in 24 listeners who had been previously tested and revealed a broad range of results (0–100%), with the mean being $68.8 \pm 34\%$. In two women with moderately severe hearing loss, the difference between $MSRS_Q$ and binaural intelligibility of monosyllabic words divided intohalves (BFT, $\Delta$) exceeded the norm, being 25% and 40%. The BFT assesses the effectiveness of acoustic information fusion from the ears. Such a fusion is considered to occur in the brainstem at the level of the superior olivary complex; thus, these tests are sensitive to the brainstem pathology [35]. However, the results of the majority of such tests are known to deteriorate in the presence of cortical disorders [25].

Dichotic tests are among the most popular methods to diagnose CAPD [25,35]. The dichotic digits test (DDT), used in this study, is relatively resistant to the influence of peripheral hearing loss [36]. Binaural integration was assessed; the patients were instructed to repeat both numbers presented to their left and right ears in an arbitrary manner (free answer). This test was the most difficult for the patients because it required high concentration. Poor results in the DDT were revealed in 87 (54%) of 161 patients, and in 14 (58%) of 24 patients who underwent expanded speech testing. Dichotic tests are proved to be highly sensitive to disorders of the auditory cortex and the corpus callosum, and they are less sensitive to disorders at the brainstem level [25]. There are data on the influence of the left frontal cortex on the dichotic testresults [37].

Impaired speech intelligibility was one of the most frequent auditory complaints in patients after COVID-19. Cognitive abilities, along with peripheral hearing and central auditory processing, are known to influence speech intelligibility [25,35]. The MoCAresults were significantly worse in the patients examined in comparison with the normative results. In 114 (71%) of 161 patients, the MoCA scale values were fewer than 26 points, indicating cognitive impairment. Moderate correlations between the MoCA values and speech intelligibility in the RuMatrix test in noise ($0.1 < p < 0.05$) and quiet ($p < 0.2$), as well as with DDT ($0.1 < p < 0.05$), were revealed. The interconnection between brain lesions, cognitive decline, and decreased speech intelligibility with the presence of CAPD was shown in various studies, including in patients with multiple sclerosis and other pathologies [38,39]. It is well known that hearing-impaired people may have decreased cognitive functions. In the current research, the MoCA test was not run before COVID-19, so we could not compare the cognitive status of patients before and after COVID-19. However, a moderate negative correlation between the MoCA results and the duration of COVID-19 ($0.1 < p < 0.05$) and a weak correlation between the MoCA results and the

severity of COVID-19 ($p < 0.2$) were found. These correlations may indicate the association between COVID-19and cognitive decline.

Mechanisms of the virus-induced central nervous system potential damage, including that of the auditory pathways and centers, are still understudied. SARS-CoV-2 was proven to spread through the olfactory epithelium and the ethmoid bone to the olfactory bulbs in the brain, causing olfactory disorders along with memory impairment and cognitive disabilities [19,22,29,40]. The central auditory pathways are suggested to be damaged by inflammation, leading to CAPD in the future.

The other possible mechanism of CAPD development may be connected with the high incidence of thrombosis caused by SARS-CoV-2. Blood coagulation and thrombosis, as its consequences, are the reasons for transitory ischemia [17]. Such a mechanism may underlie both peripheral and central auditory disorders depending on clot location. The lack of a significant correlation between the severity of the disease and the presence of auditory disorders may be explained by the use of anticoagulants in severe cases. The same mechanism may be the cause of vertigo and tinnitus in patients after recovery from COVID-19.

## 5. Limitations

The cognitive function was not evaluated in patients before COVID-19. Despite a large number of patients examined, the group was rather heterogeneous—patients differed by hearing functions and age, and most of the patients (85%) did not have any audiometry data before COVID-19. All these factors limit the sample size and may affect the results of the CAPD detection tests.

## 6. Conclusions

Decreased results of speech tests in patients after COVID-19 may be a sign of central auditory processing disorders, as well as a consequence of impaired memory and deteriorated cognitive abilities. Further audiology monitoring of patients after COVID-19 is needed to evaluate the potential influence of the virus on the auditory system in the long term.

**Author Contributions:** Conceptualization: M.Y.B., E.S.G. and L.E.G.; methodology: M.Y.B., E.S.G., S.M.V. and L.E.G.; formal analysis: E.A.O., M.Y.B., E.S.G. and E.V.Z.; investigation: M.Y.B., E.S.G., S.M.V., L.E.G. and A.V.R.; resources: M.Y.B. and L.E.G.; data curation: S.M.V.; writing—original draft preparation: M.Y.B., E.S.G., S.M.V., L.E.G. and E.A.O.; writing—review and editing: M.Y.B., E.S.G., S.M.V., E.A.O. and E.V.Z.; supervision: M.Y.B.; project administration: M.Y.B. All authors have read and agreed to the published version of the manuscript.

**Funding:** This research received no external funding.

**Institutional Review Board Statement:** The study was conducted according to the guidelines of the Declaration of Helsinki and approved by the Bioethics Committee of the Saint-Petersburg Geriatric Medico-social Center (number: IFPS: 8/04/2021) and by the Bioethics Committee of Pavlov First St. Petersburg State Medical University (number: IFPS: 261).

**Informed Consent Statement:** Informed consent was obtained from all subjects involved in the study.

**Data Availability Statement:** The data presented in this study are available on request from the corresponding author. The data are not publicly available due to protection of personal medical data.

**Acknowledgments:** The authors would like to express their gratitude to the staff of the Laboratory of Hearing and Speech of Pavlov First St. Petersburg State Medical University and Municipal Audiology Department of Saint-Petersburg Geriatric Medico-social Center for assistance in conducting audiological examination.

**Conflicts of Interest:** The authors declare no conflict of interest.

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
