# Peer review of "The New Coronavirus Infection (COVID-19) and Hearing Function in Adults"

_2504-463X, doi:10.3390/ohbm3020005_

Round 1

Reviewer 1 Report

The authors present an interesting observational study examining subjects hearing and central auditory processing disorder (CAPD) in subjects pre and post COVID. The authors examine 161 subjects who had audiological examinations post-COVID; of these, 24 had audiological examinations pre- and post-COVID. Many existing studies pertaining to hearing and COVID investigate sudden sensorineural hearing loss; CAPD is less investigated. The methods are strong and the statistical analysis robust. Findings suggest a potential effect of COVID on central auditory processing. 

However, there are some questions that can be further addressed in the manuscript that will make it even stronger.

1) The authors' results showed that the subjects who underwent MoCA, RGDT, MSRSq, MSRSn, dichotic digits tests had results that were below that of the normative values. Most of the subjects seemed to be recruited from audiologists (85% were primary patients of audiologists). Subjects who present to an audiology clinic with hearing complaints may be more likely to have decreased cognitive function compared to a normal hearing individual—there is a known association between hearing loss and cognitive decline. As such, their scores may be their baseline and not a result of COVID, since pre and post-COVID testing of CAPD was not performed. Although the authors did perform correlation analyses that suggested that duration of the disease was associated with poorer performance on these tests, the results are underpowered given the limited sample size, this is a significant limitation and should be addressed in the manuscript. 

2) The authors should have somebody check the English of the manuscript - there are some errors (e.g., "Despite speech audiometry in quiet doesn't relate to the tests for CAPD detection, low scores of speech intelligibility in quiet...")

Author Response

Dear Reviewer,

We are very grateful for the work you have done and for your crucial comments. We have taken into account all your recommendations.

We totally agree with your comments. According them we changed the sections Discussion and Limitation.

All our corrections are highlighted blue in the text attached.

In order to follow your recommendation to improve English, we will definitely use the service of MDPI.

Thank you!

Sincerely yours

Reviewer 2 Report

Dear authors 

The topic is good to study, whoever, it's a too lengthy paper with 64 references, most of the journals for research paper accepts 30-40references which would be ideal. I feel there is a need to condense the manuscript

Author Response

Dear Reviewer,

We are very grateful for the work you have done and for your crucial comment.

In order to follow your recommendation we have reduced the reference list to 40.

All our corrections are highlighted blue in the text attached.

Thank you!

Sincerely yours

Reviewer 3 Report

Line 37: Remove words, "of the II pathogenicity group."

Line 38: "There are following routes of COVID-19", change this to "These are the following..."

Line 79: It is not ideal to use words such as "and/or" in research papers. 

-Overall an interesting article that brings up an important point to discuss the impact of COVID-19 on hearing. The article can use english language editing but overall well written and easy to follow. 

Author Response

Dear Reviewer,

We are very grateful for the work you have done and for your crucial comments.

We have made corrections in the Line 37, 38, 79.

All our corrections are highlighted blue in the text attached.

In order to follow your recommendation to improve English, we will definitely use the service of MDPI.

Thank you!

Sincerely yours

Round 2

Reviewer 2 Report

Dear authors, i appreciate the changes made in the manuscript, i have no further comments.

Reviewer 3 Report

The authors have made appropriate changes in the article as suggested by reviewers. The article should get published once extensive english language editing is performed.